# The Association between Plasma ERVWE1 Concentrations and Affective Symptoms during Pregnancy: Is This a Friendly Alien?

**DOI:** 10.3390/ijerph17249217

**Published:** 2020-12-09

**Authors:** Marta Serati, Cecilia Maria Esposito, Silvia Grassi, Valentina Bollati, Jennifer Lynn Barkin, Massimiliano Buoli

**Affiliations:** 1Department of Mental Health, ASST RHODENSE, 20024 Rho, Italy; martaserati@libero.it; 2Department of Pathophysiology and Transplantation, University of Milan, 20122 Milan, Italy; cecilia.esposito@unimi.it (C.M.E.); silvia.grassi1@unimi.it (S.G.); 3EPIGET-Epidemiology, Epigenetics and Toxicology Lab-Department of Clinical Sciences and Community Health, University of Milan, 20122 Milan, Italy; valentina.bollati@unimi.it; 4Department of Community Medicine, Mercer University School of Medicine, Macon, GA 31207, USA; barkin_jl@mercer.edu; 5Department of Neurosciences and Mental Health, Fondazione IRCCS Ca’Granda Ospedale Maggiore Policlinico, Via F. Sforza 35, 20122 Milan, Italy

**Keywords:** Human Endogenous Retrovirus W EnvC7-1 Envelope Protein (ERVWE1), syncytin 1, pregnancy, affective symptoms

## Abstract

Human endogenous retroviruses (HERVs) comprise 8% of the human genome, and HERV DNA was reported to be essential in human embryonic development. Specifically, HERV-W encodes a protein, syncytin-1, alternatively known as ERVWE1 (Human Endogenous Retrovirus W EnvC7-1 Envelope Protein), participating in human placental morphogenesis and having a role in immune system regulation. Syncytin-1 activity is increased in neuropsychiatric disorders, autoimmune diseases, and cancer. In our study, forty-four women in the third trimester of pregnancy were tested for ERVWE1 plasma levels. In concomitance with blood samples the following rating scales were administered to women: the Edinburgh Postnatal Depression Scale (EPDS), State Anxiety Inventory (STAI-S), Trait Anxiety Inventory (STAI-T), and Prenatal Attachment Inventory (PAI). We found that higher ERVWE1 protein plasma levels were significantly associated with higher PAI scores (*p* = 0.02), an earlier gestational age at the time of blood collection (*p* = 0.01), a longer duration of symptoms (*p* = 0.03), and fewer lifetime attempted suicides (*p* = 0.02). Our results seem to support the role of ERVWE1 in maintaining clinical psychiatric symptoms as a result of potential prolonged inflammation. At the same time, this protein may have a protective role in pregnant women by a reduction of suicidal behavior and a better mother–fetus relationship.

## 1. Introduction

It is well established that affective disorders have a multifactorial etiology. Specifically, inflammation can trigger depressive symptoms onset with subsequent negative obstetric and neuropsychiatric outcomes [1]. Human endogenous retroviruses (HERVs) are actually considered to be potential pathogenetic factors, triggering the immune system in a variety of ways. Their role is under study in the context of neurological and psychiatric disorders [2,3,4].

HERVs comprise three main classes: class I (HERV-W, HERV- H), class II (HERV-K), and class III (HERV-L). This study was focused on syncytin-1, the envelope protein encoded by HERV-W, alternatively known as ERVWE1 (Human Endogenous Retrovirus W EnvC7-1 Envelope Protein). This protein acts as an immunotoxin, induces inflammation, activates the innate immune system, and upregulates proinflammatory cytokines (e.g., interleukin 1-IL-1, interleukin 6-IL-6). Of note, increased ERVWE1 activity was found in some inflammatory diseases such as multiple sclerosis [5,6]. Despite the role of ERVWE1 in triggering inflammatory processes, the results of some studies indicate that this molecule may modulate the immune system in some specific situations such as pregnancy [7].

With regard to psychiatric conditions, increased expression of Human Endogenous Retrovirus-W (HERV-W) was found in liquor of patients affected by schizophrenia [8]. In addition, Weis and colleagues [9] reported an alteration in glial expression of proteins codified by the HERV-W gene in patients affected by severe psychiatric disorders versus healthy controls. Finally, a recent study reported a trend of HERVs increased expression both in patients with schizophrenia and bipolar disorder [10].

Syncytin-1 (ERVWE1) expression during pregnancy is regulated by epigenetic and nonepigenetic mechanisms, such as progesterone and proinflammatory cytokine levels (Tumor Necrosis Factor α-TNF-α, Interferon γ-IFN-γ, and IL-6). ERVWE1 plays an important role in normal syncytium formation and its maintenance, being a key mediator of cytotrophoblasts fusion process [11,12,13]. During pregnancy, cellular-mediated immunity has to be suppressed, and ERVWE1 contributes to maternal-fetal tolerance [14,15], inhibiting Th1 cytokines (TNF-α, IFN-γ, and interleukin 2-IL-2) and promoting the shift to Th2-mediated immunity [15]. Moreover, the promoter hypomethylation of the corresponding gene is essential for pregnancy maintenance by organizing placenta development and perhaps, in humans, by modulating progesterone action, thus affecting gestational length and labor onset [14,16].

Two studies reported that primate endogenous retroviruses are placenta-specific enhancers of corticotropin-releasing hormone (CRH) that acts in birth timing control [17,18]. Of note, HERV over-expression was reported to be associated with an early gestational age [19]. In contrast, hyper-methylation of the ERVWE1 gene promoter was found in placentas of women affected by pre-eclampsia compared to healthy controls [20]. In agreement, reduced syncytin-1 (ERVWE1) expression in trophoblastic lineages was reported in placentas of women with pre-eclampsia compared to controls [13]. A study on syncytins and their receptors during normal placental development demonstrated the importance of these proteins for normal placenta functioning [21].

Taken as a whole, while human endogenous retroviruses (HERVs) seem to have a negative effect on human health, the envelope protein (ERVWE1) appears to have a role in the maintenance of pregnancy. For this reason, the purpose of the present study is to investigate the association between women’s mental health during pregnancy and ERVWE1 plasma levels.

## 2. Materials and Methods

### 2.1. Participants

Forty-four women in the third trimester of pregnancy were selected for the study. The sample consisted of healthy pregnant women (n = 14) and women affected by perinatal major depressive disorder (PMDD) (n = 30), and it was monitored by consultation psychiatric service. Exclusion criteria included the following: (1) being less than 18 years, (2) the incapacity to express informed consent, (3) current treatment with drugs (e.g., corticosteroids), or (4) medical comorbidities (e.g., autoimmune diseases) that can potentially impact both affective symptom severity and ERVWE1 plasma levels [15,22]. If we assume an average difference of Edinburgh Postnatal Depression Scale (EPDS) scores between our sample (mixed healthy and depressed pregnant women) and healthy pregnant women of 3 points (standard deviation ± 5), and if a value of *p* = 0.05 is considered significant, for a power of 90% a sample of 30 is calculated as adequate.

### 2.2. Procedures

All participants signed an informed consent, and study procedures were approved by the local Medical Ethics Review Committee. A blood sample was collected in the late morning just before the assessment of mental health. ERVWE1 plasma levels were measured with an ELISA kit (CliniSciences, Nanterre, France). The laboratory staff was blind with respect to the women’s mental health status.

### 2.3. Measures

The severity of anxiety and depressive symptoms was measured by the EPDS [23], the State Anxiety Inventory (STAI-S), and the Trait Anxiety Inventory (STAI-T) [24]. The EPDS is an instrument with high test–retest reliability, commonly used to monitor depressive symptoms during the perinatal period [25]. STAI-S and STAI-T are sensitive in measuring the level of anxiety of both healthy subjects and those affected by anxiety disorders [24]. The quality of the relationship between mother and fetus was evaluated by the Prenatal Attachment Inventory (PAI) [26]. Duration of untreated symptoms was defined as the time between the onset of anxiety or depressive symptoms and the start of a recommended treatment according to guidelines (antidepressants or cognitive psychotherapy) [27,28].

### 2.4. Statistical Analyses

Descriptive analyses of the total sample were performed. Mean ERVWE1 plasma levels were compared between healthy women and those with PMDD by one-way analysis of variance. The association between demographic and clinical factors (independent variables) and ERVWE1 plasma levels (dependent variable) was analyzed by performing a multivariate logistic regression. SPSS version 26 was used as statistical software.

## 3. Results

ERVWE1 plasma levels were measured in 44 pregnant women with a mean age of 36.24 ± 4.42 years. Descriptive analyses of the total sample are reported in Table 1.

No differences were found between women affected by PMDD (43.05 ± 3.83 ng/mL) and healthy controls (43.65 ± 3.40 ng/mL) (F = 0.23, *p* = 0.64).

The resultant multivariate linear regression model proved reliable (Durbin-Watson: 1.75). In our sample, ERVWE1 protein plasma levels were significantly associated with higher PAI scores (β = 0.76, *p* = 0.018), an earlier gestational age at the time of blood sample collection (β = −0.80, *p* = 0.015), a longer duration of symptoms (β = 1.03, *p* = 0.029), and fewer lifetime attempted suicides (β = −1.37, *p* = 0.021) (Table 2).

## 4. Discussion

Syncytin-1 (ERVWE1) induces cell–cell fusion, as demonstrated in vitro, playing an important role in placentation. Syncytin-1 is fundamental for favorable pregnancy outcomes because it blocks immune response against the fetus by creating an immunologic barrier. Of note, this protein is upregulated by progesterone. In our study, the diagnosis of PMDD does not seem to influence the protein plasma levels. In contrast, higher ERVWE1 protein plasma levels were significantly associated with higher PAI scores, earlier gestational age at the time of blood collection, a longer duration of symptoms, and fewer lifetime attempted suicides. Our data seem to indicate a protective role in pregnancy for syncytin-1, in particular with regard to suicidal behavior and mother–fetus relationship, as shown by the higher ERVWE1 plasma levels in the case of high PAI scores. A previous study from our team, conducted in women with perinatal depression, found that high PAI scores seem to be associated with a favorable pattern of clock genes methylation. In particular, we found a lower methylation of *CLOCK* (expected to be higher in pregnant women, especially the depressed ones) and *HERV-W* genes (essential for a favorable pregnancy outcome) in women with the highest PAI scores [29].

Conversely, our findings seem to support the possible role of ERVWE1 in the maintenance of poor mental health as shown by the direct association between protein plasma levels and duration of affective symptoms in our sample. This association might be explained by the role of ERVWE1 in enhancing inflammation. Of note, in multiple sclerosis ERVWE1 was reported to act as a trigger of relapse by activation of leukocytes and cytokine cascade [6]. As reported by Garcia Montojo and colleagues [6], syncytin-1 (ERVWE1) could favor cell-to-cell contact to form the immunological synapse between the antigen-presenting cells and the lymphocytes with a consequent immune activation. These authors found that syncytin-1 is intrinsically upregulated in multiple sclerosis, inducing a preactivated status of different leucocytes and determining proinflammatory cytokine release. In agreement with these results, statistically significant correlations were reported between HERV-W levels and diagnosis of multiple sclerosis [30].

Finally, during early stages of pregnancy ERVWE1 seems to be upregulated, whereas a downregulation happens with the progression of pregnancy, in agreement with the results of our study that found an inverse association between gestational age and protein plasma levels. It will be of interest to know how ERVWE1 changes in relation to the mother’s psychopathological status in larger samples to confirm the results of the present study. In addition, future research should clarify the effect of ERVWE1 exposure on the neurodevelopment of the offspring.

Limits of the study consisted of the small sample size, the risk of type 1 statistical error, and the potential influence of antidepressant treatment on ERVWE1 plasma levels, although only 8 out of 30 depressed women received pharmacotherapy at the time of the blood sample (25 mg of citalopram: n = 2; 50 mg of sertraline: n = 3; 20 mg of paroxetine: n = 3). Finally, given the cross-sectional nature of the study, no causal relationship can be made.

## 5. Conclusions

This is one of the first studies investigating how ERVWE1 plasma levels can affect mental health of women during pregnancy. The diagnosis of PMDD does not seem to influence the protein plasma levels. In contrast, while higher plasma levels of this protein might be related to poor mental health in pregnant women in terms of duration of symptoms, on the other hand it might favor the maintenance of pregnancy by preventing suicidal attempts and by enhancing mother–fetus relationship. We hypothesize that, on the one hand, the affective symptoms can maintain a certain inflammatory state, but on the other hand this protein can play an important role in safeguarding the fetus by mitigating the suicidal risk and improving the mother–fetus relationship. The function of this protein in humans is controversial, and further studies are required to have data about the variations of the plasma levels of this protein in the general population to define the effects of pregnancy [31]. Of note, few data were published about the ERWE1 serum/plasma levels in the general healthy population as well as in patients with psychiatric disorders. In a study comparing healthy controls and schizophrenia subjects, mean ERWE1 values of 17.21 ± 16.61 ng/mL for healthy volunteers and of 72.47 ± 69.07 for patients were found [32]. Finally, future studies should also investigate the effect of exposition to this protein during pregnancy on offspring neurodevelopment.

## Figures and Tables

**Table 1 ijerph-17-09217-t001:** Descriptive analyses of the total sample.

Quantitative Variables	Mean	SD
ERVWE1 plasma levels (ng/mL)	43.70	2.79
Age	36.24	4.42
Number of attempted suicides	1.14	0.36
Duration of untreated symptoms (months) *	2.52	3.72
Gestational age at the time of blood sample collection (weeks)	32.90	4.18
Number of previous pregnancies	1.43	0.81
EPDS scores	10.29	9.43
STAI-T scores	49.05	14.47
STAI-S scores	51.43	16.77
PAI scores	56.90	9.26
Duration of symptoms (months) *	11.81	29.19
Gestational age at birth (weeks)	39.05	1.28
Birth weight (grams)	3246.90	436.76

Legend: EPDS: Edinburgh Postnatal Depression Scale; ERVWE1: Human Endogenous Retrovirus; W EnvC7-1 Envelope Protein; PAI: Prenatal Attachment Inventory; SD: Standard deviation; STAI-S: State Anxiety Inventory; STAI-T: Trait Anxiety Inventory; * For healthy controls these variables were computed as = 0.

**Table 2 ijerph-17-09217-t002:** Multivariate linear logistic regression model about the association between demographic/clinical variables and plasma Human Endogenous Retrovirus W EnvC7-1 Envelope Protein (ERVWE1) concentrations.

Variables	ß	CI	*p*
Age	−0.36	−0.56/0.10	0.14
Number of attempted suicides	−1.37	−19.29/−2.10	**0.02**
Duration of untreated symptoms (months)	−0.41	−0.90/0.29	0.27
Gestational age at the time of blood sample collection (weeks)	−0.80	−0.94/−0.13	**0.01**
Number of previous pregnancies	0.36	−1.07/3.55	0.25
EPDS scores	0.04	−0.49/0.52	0.95
STAI-T scores	0.05	−0.17/0.19	0.90
STAI-S scores	1.03	−0.04/0.38	0.09
PAI scores	0.76	0.05/0.41	**0.02**
Duration of symptoms (months)	1.03	0.01/0.18	**0.03**
Gestational age at birth (weeks)	0.12	−1.07/1.59	0.67
Child birth weight (grams)	−0.37	−0.01/0.002	0.20

In bold, statistically significant *p* values; Legend: ß: Standardized coefficient; CI: Confidence interval; EPDS: Edinburgh Postnatal Depression Scale; ERVWE1: Human Endogenous Retrovirus W EnvC7-1 Envelope Protein; PAI: Prenatal Attachment Inventory; SD: Standard deviation; STAI-S: State Anxiety Inventory; STAI-T: Trait Anxiety Inventory.

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
