# Peer review of "The Association between Plasma ERVWE1 Concentrations and Affective Symptoms during Pregnancy: Is This a Friendly Alien?"

_ijerph, 2020, doi:10.3390/ijerph17249217_

Round 1
Reviewer 1 Report
The idea proposed in your work is interesting and novel, but several methodological issues need to be clarified.
- Inclusion criteria are not mentioned and it is not clear the neuropsychological profile of the 44 enrolled patients
- A control group is not esplicited. In comparison to what, you declare that in your sample of patients ERVWE1 plasma levels are increased? (line 111)
- You have to explain how you calculated your sample size of 44 women (materials and methods section 2.1 Participants)
- Which software did you use for your statistical analysis?
- In the discussion section you have to highlight the limitations of your study as well.
- In the discussion section you also should propose a possible explanation to why increased ERVWE1 plasma levels are associated with poor maternal health status and at the same time with reduction of suicidal behavior and a better mother-fetus relationship. You esplicited that this result is conflicting, but you should propose a possible explanation.
Author Response
First of all, we would like to thank the first Reviewer for the interest in the manuscript and the useful comments aimed to improve the present manuscript.
The idea proposed in your work is interesting and novel, but several methodological issues need to be clarified.
Thanks for the appreciation.
1) Inclusion criteria are not mentioned and it is not clear the neuropsychological profile of the 44 enrolled patients
Thanks for your observation. We did not mention this information for short report format of the publication. The sample consisted of healthy pregnant women (n=14), or women affected by perinatal major depressive disorder (n=30) and monitored by consultation psychiatric service. This information has been reported in materials and methods.
2) A control group is not esplicited. In comparison to what, you declare that in your sample of patients ERVWE1 plasma levels are increased? (line 111)
We meant that higher ERVWE1 plasma levels were directly associated with higher PAI scores, an earlier gestational age at the time of blood sample collection, a longer duration of symptoms and less lifetime attempted suicides. We removed “increased” to avoid ambiguity throughout the manuscript. We reported also the results of the comparison of the protein plasma levels between women with perinatal depression versus healthy controls.
3) You have to explain how you calculated your sample size of 44 women (materials and methods section 2.1 Participants)
If we assume an average difference of EPDS scores between our sample (mixed healthy and depressed pregnant women) and healthy pregnant women of 3 points (standard deviation ± 5) and if a value of p= 0.05 is considered significant, for a power of 90% a sample of 30 is calculated as adequate. We reported this information in the section about participants.
4) Which software did you use for your statistical analysis?
We used SPSS version 26. We added this information in the part about statistical analyses.
5) In the discussion section you have to highlight the limitations of your study as well.
We added the study limitations (small sample size, treatment with antidepressant, cross-sectional nature of the study) according to your suggestion
6) In the discussion section you also should propose a possible explanation to why increased ERVWE1 plasma levels are associated with poor maternal health status and at the same time with reduction of suicidal behavior and a better mother-fetus relationship. You esplicited that this result is conflicting, but you should propose a possible explanation.
Thanks for your useful comments. We hypothesize that on the one hand the affective symptoms can maintain a certain inflammatory state, but on the other hand this protein can play an important role in safeguarding the fetus by mitigating the suicidal risk and improving the mother-fetus relationship.
Reviewer 2 Report
This is a highly original study looking at association of ERVWE1 plasma levels and affective symptoms during pregnancy. The title is quite eye catching (friendly alien?).
It is important throughout the paper to distinguish between associations and causation. The statement for example in the conclusion that "it seems to have a beneficial effect...by preventing suicidal attempts and by enhancing mother-fetus relationship" suggests causation. It would be fairer to state that this association is interesting and might suggest a causal relationship.
I would expect to see an impact on depression scores if indeed there was causal relationship in preventing suicide. The authors did not see this although they used EPDS which is a postnatal depression score. They may like to explain this discrepancy.
Also the number of attempted suicides was very low and the association might therefore be a type 1 statistical error.
It is not clear how these women were selected. What proportion of them had affective symptoms? Are they representative of the general population of women attending an antenatal clinic? The mean duration of affective symptoms of 11.8 months implies that these pre-date the pregnancy so the associations might not relate to pregnancy.
Do we have a reference range for plasma ERVWE1 in pregnant women? Were the levels seen outside of the reference range. The relation with gestational age of the women (all were in the third trimester) might simply reflect the normal variation towards the end of pregnancy.
Line 50 "autoimmune diseases such as multiple sclerosis" - I am not aware that multiple sclerosis has been definitely classified as an autoimmune disease.
Line 65 "hypometilation" - hypomethylation
Author Response
First of all, we would like to thank the second Reviewer for the interest in the manuscript and the helpful comments aimed to improve the present manuscript.
This is a highly original study looking at association of ERVWE1 plasma levels and affective symptoms during pregnancy. The title is quite eye catching (friendly alien?).
Thanks for the appreciation of the manuscript.
1) It is important throughout the paper to distinguish between associations and causation. The statement for example in the conclusion that "it seems to have a beneficial effect...by preventing suicidal attempts and by enhancing mother-fetus relationship" suggests causation. It would be fairer to state that this association is interesting and might suggest a causal relationship.
Thanks for the suggestion. We mitigated the sentence and we specified in the limits of the study that, given the cross-sectional nature of the study no causal relationship can be made.
2) I would expect to see an impact on depression scores if indeed there was causal relationship in preventing suicide. The authors did not see this although they used EPDS which is a postnatal depression score. They may like to explain this discrepancy.
As mentioned above, we specified in the limits of the study that, given the cross-sectional nature of the study no causal relationship can be made. In addition, the diagnosis of Perinatal Major Depressive Disorder does not seem to change significantly the plasma levels of the protein. We reported these data in the results and we commented them. We also detailed a hypothesis about the “ambivalent role” of this protein in pregnant women. Clearly only future research will be able to confirm or disconfirm these findings.
3) Also the number of attempted suicides was very low and the association might therefore be a type 1 statistical error.
We added the risk of this type of error as a study limitation, however we specified the sample size calculation in the section about participants.
4) It is not clear how these women were selected. What proportion of them had affective symptoms? Are they representative of the general population of women attending an antenatal clinic? The mean duration of affective symptoms of 11.8 months implies that these pre-date the pregnancy so the associations might not relate to pregnancy.
The sample consisted of 30 women affected by Perinatal Major Depressive Disorder and 14 healthy controls. The depressed patients were monitored by consultation psychiatric service. The third observation is intriguing and it can be a topic for future research. As we reported in the conclusions, we hypothesize that on the one hand the affective symptoms can maintain a certain inflammatory state, but on the other hand this protein can play an important role in safeguarding the fetus by mitigating the suicidal risk and improving the mother-fetus relationship. We specified that further studies are required to have data about the variations of the plasma levels of this protein in the general population to have a more precise idea of the effects of pregnancy.
5) Do we have a reference range for plasma ERVWE1 in pregnant women? Were the levels seen outside of the reference range. The relation with gestational age of the women (all were in the third trimester) might simply reflect the normal variation towards the end of pregnancy.
Few data were published about the ERWE1 serum/plasma levels. In a study (doi: 10.1016/j.bbi.2017.09.009) comparing healthy controls and schizophrenia subjects was found a mean value of ERWE1 of 17.21 ± 16.61 ng/mL for healthy volunteers and of 72.47 ± 69.07 for patients. We reported these information in the conclusions. Our sample seems to have an intermediate value and it could be hypothesized that affective symptoms collocate this protein in an intermediate level. ERWE1 is basically more expressed in the first trimester of pregnancy and progressively decline with the advancement of pregnancy being implicated in the process of placentation. It is not therefore surprising that also in the context of the third trimester there is an inverse relation of the gestational age with the ERWE1 plasma levels (doi: 10.1016/j.placenta.2012.02.012)
6) Line 50 "autoimmune diseases such as multiple sclerosis" - I am not aware that multiple sclerosis has been definitely classified as an autoimmune disease.
Basically it is considered a neurological inflammatory autoimmune disease (e.g. doi: 10.1016/B978-0-12-804279-3.00005-8). We substituted the term “autoimmune” with “inflammatory” to avoid ambiguity.
7) Line 65 "hypometilation" - hypomethylation
Thanks we corrected it.
Round 2
Reviewer 1 Report
Dear authors,
Thank you for your reply. I don't think that further changes need to be made to the manuscript. It is of course reccomended to keep going with your research in the future to add useful elements to explain your interesting findings.
In the meantime, congratulations for your work.
Reviewer 2 Report
The authors have made appropriate changes to the paper which is now suitable for publication